# Synthesis of Novel Acyl Derivatives of 3-(4,5,6,7-Tetrabromo-1*H*-benzimidazol-1-yl)propan-1-ols—Intracellular TBBi-Based CK2 Inhibitors with Proapoptotic Properties

**DOI:** 10.3390/ijms22126261

**Published:** 2021-06-10

**Authors:** Konrad Chojnacki, Patrycja Wińska, Olena Karatsai, Mirosława Koronkiewicz, Małgorzata Milner-Krawczyk, Monika Wielechowska, Maria Jolanta Rędowicz, Maria Bretner, Paweł Borowiecki

**Affiliations:** 1Chair of Drug and Cosmetics Biotechnology, Faculty of Chemistry, Warsaw University of Technology, 00-664 Warsaw, Poland; kchojnacki@ch.pw.edu.pl (K.C.); mmilnerkrawczyk@ch.pw.edu.pl (M.M.-K.); mwielechowska@ch.pw.edu.pl (M.W.); mabretner@gmail.com (M.B.); Pawel.Borowiecki@pw.edu.pl (P.B.); 2Laboratory of Molecular Basis of Cell Motility, Nencki Institute of Experimental Biology, Polish Academy of Sciences, 02-093 Warsaw, Poland; o.karatsai@nencki.edu.pl (O.K.); j.redowicz@nencki.edu.pl (M.J.R.); 3Department of Drug Biotechnology and Bioinformatics, National Medicines Institute, 00-725 Warsaw, Poland; m.koronkiewicz@nil.gov.pl

**Keywords:** protein kinase CK2, CK2 inhibitors, breast cancer, tetrabromobenzimidazole, apoptosis, prodrug strategy

## Abstract

Protein kinase CK2 has been considered as an attractive drug target for anti-cancer therapy. The synthesis of *N*-hydroxypropyl TBBi and 2MeTBBi derivatives as well as their respective esters was carried out by using chemoenzymatic methods. Concomitantly with kinetic studies toward recombinant CK2, the influence of the obtained compounds on the viability of two human breast carcinoma cell lines (MCF-7 and MDA-MB-231) was evaluated using MTT assay. Additionally, an intracellular inhibition of CK2 as well as an induction of apoptosis in the examined cells after the treatment with the most active compounds were studied by Western blot analysis, phase-contrast microscopy and flow cytometry method. The results of the MTT test revealed potent cytotoxic activities for most of the newly synthesized compounds (EC_50_ 4.90 to 32.77 µM), corresponding to their solubility in biological media. We concluded that derivatives with the methyl group decrease the viability of both cell lines more efficiently than their non-methylated analogs. Furthermore, inhibition of CK2 in breast cancer cells treated with the tested compounds at the concentrations equal to their EC_50_ values correlates well with their lipophilicity since derivatives with higher values of log*P* are more potent intracellular inhibitors of CK2 with better proapoptotic properties than their parental hydroxyl compounds.

## 1. Introduction

Breast cancer is the most prevalent malignancy in females and the second most common cause of death in women after lung cancer [1]. Surgical treatment as well as chemo-, hormone-, or antibodies-based therapies are currently used, depending on the type of breast cancer. Based on the presence of endocrine receptors such as estrogen receptors (ER), progesterone (PR) and epidermal growth factor 2 (HER2), there are several biological types of breast cancer that differ in their sensitivity/resistance towards the applied therapy [2]. Triple negative breast cancer (TNBC), the most difficult type of breast carcinoma to treat, accounting for 10 to 20% of all invasive breast cancers [3], is negative for the HER2 amplification, and both progesterone receptor (PR) and estrogen receptor (ER) expression [4]. Since TNBC is resistant to current hormone-based chemotherapies or anti-HER2 treatments [5,6], there is still an urgent need to search for a new molecular targets and therapeutic approaches that would be effective in TNBC treatment [7,8]. Protein kinase CK2 (formerly known as casein kinase II) is an enzyme that catalyzes phosphorylation of a huge number of substrates, and thus is involved in the regulation of many processes such as transcription [9,10], translation [11,12,13,14], control of protein stability [15,16,17] and degradation [18,19], cell cycle progression [20], cell survival [21,22,23] as well as circadian rhythms [24]. The role of this conserved constitutively active serine-threonine kinase in the cell regulatory network is highly complex, and the extensive interplay between CK2-mediated phosphorylation and other post-translational modifications has been suggested [25]. The analysis of phosphoproteomic datasets typically suggests that CK2 could be responsible for more than 10% of the phosphoproteome [26]. Most of the CK2 substrates play important roles in the regulation of metabolic processes, cell signaling and apoptosis [27,28]. Moreover, an increased level/activity of CK2 kinase has been observed in several tumor types [29], including breast cancer, where overexpression of CK2 results in the dysregulation of key cellular signalling pathways that control transcription factors of mammary epithelium [30]. Increased catalytic activity of CK2 has also been linked to Her-2/neu oncogene, which is overexpressed in 30% of breast cancers [31]. Furthermore, the elevated CK2 activity is a key factor in the promotion of the Her-2/neu survival pathways leading to higher tumor growth rates in vivo [32]. Many studies demonstrated that inhibition of CK2 by small ATP-competitive inhibitors, such as 4,5,6,7-tetrabromo-1*H*-benzimidazole (TBBi) and its derivatives, or by silmitasertib (code name: CX-4945) (Figure 1), is responsible for a range of phenotypic changes in breast cancer cell lines, including decreased viability, cell cycle arrest, apoptosis, and loss of migratory capacity [33]. 

Among several classes of effective CK2 inhibitors are derivatives of benzotriazole and benzimidazole, i.e., 4,5,6,7-tetrabromo-1*H*-benzotriazole (TBBt), 4,5,6,7-tetrabromo-1*H*-benzimidazole (TBBi) and 4,5,6,7-tetrabromo-*N*,*N*-dimethyl-1*H*-benzimidazol-2-amine (DMAT) (Figure 1). We previously found that attachment of the methyl group to the *C*2 atom of TBBi and the aminoalkyl-like substituents at the *N*1 atom led to the 4,5,6,7-tetrabromo-2-methyl-1*H*-benzimidazole (2MeTBBi) (Figure 1) and their *N*-aminoalkyl derivatives with a promising inhibitory activity towards CK2 and increased cytotoxicity against both CCRF-CEM and MCF-7 cell lines [34]. Another TBBi derivative that effectively inhibits the activity of CK2 is 3-(4,5,6,7-tetrabromo-1*H*-benzimidazol-1-yl)propan-1-ol (*K*_i_ 0.28 µM), which also inhibits the proliferation of CCRF-CEM (EC_50_ = 13.5 µM) and MCF-7 (EC_50_ = 15.4 µM) tumor cells [35]. 

However, a previous report showed that despite the fact that *N*-hyroxypropyl TBBi, compound **2** exhibited better inhibitory activity against recombinant CK2 than parent TBBi, its inhibitory potency towards the target enzyme in cellulo was markedly lower (based on the immunodetection of p65 on Ser529 in CCRF-CEM cell line after 48-h treatment) [35]. One of the reasons for this phenomenon can be attributed to the decreased ability of these compounds to diffuse across the cell membranes. In this regard, various methods can be used to facilitate transcellular permeability, including modification of the active substances into prodrugs characterized by higher lipophilicity. Prodrugs typically refer to the bioreversible derivatives of drug molecules with no or low cytotoxicity that need to be bioconverted to an active form to exert the desired pharmacological effects. Probably, the easiest way to achieve this goal is to esterify the hydroxyl group of the designed by us inhibitors **2** and **5**. Depending on the acyl donor used, esters with different lipophilicity and in vivo lability can be obtained.

In general, in the literature, there are reports on ester prodrugs with different properties modified, which are metabolically transformed into hydroxyl-containing biologically active compounds. The most prominent examples of such a strategy and likely the most common prodrugs ever used in pharmacy are the esters of sex hormones (i.e., 17β-*O*-acylated testosterone) [36,37] and vitamins (i.e., α-tocopheryl acetate) [38]. The other prime example of derivatized therapeutic agent concerns chloramphenicol palmitate, which was designed to improve very poor pharmacokinetic profile of the original antibiotic [39]. Furthermore, famciclovir is a diacetylated (and 6-deoxy) prodrug of penciclovir with a 15-fold higher bioavailability than the parent anti-herpesvirus agent [40,41]. In turn, derivatization of antipsychotic haloperidol by means of fatty acids of different aliphatic chain lengths (up to decanoate ester) furnished excellent prodrugs with sustained release and significantly prolonged duration of action [42]. Finally, anti-cancer drug gemcitabine esterified with various aliphatic amino acids yielded prodrugs with high affinity for the PEPT1 transporter [43], whereas a powerful anti-leukemic agent, namely 9-β-D-arabinofuranosyl guanine (*ara*-G), acetylated regioselectively at 5′-position rendered derivative of significantly improved solubility and bioavailability [44,45]. Other examples of prodrug strategies for cancer cell-specific targeting, extensively utilized in oncotherapies, can be found in recent review articles [46,47]. 

In view of the above-mentioned findings, we decided to synthesize a series of fatty acid esters of *N*-hydroxypropyl-substituted TBBi derivatives, and further evaluate the influence of aliphatic chain length of carboxylic acid moiety on their anti-cancer activity. Here, we report the experimental details of the synthesis of *N*-hydroxypropyl derivatives of TBBi (**2**) and 2MeTBBi (**5**) as well as their corresponding esters (**3a**–**d**, **6a**–**d**). To gain more insight on the substitution pattern of the acyl moiety in TBBi derivatives, a structure–function relationship analysis was performed based on complex biological studies. In this context, we examined the cytotoxic effects of all the obtained esters against two different human breast cancer cell lines (i.e., MCF-7 and MDA-MB-231), differing in *p53* status and CK2 level [33]. Additionally, we evaluated the inhibition of protein kinase CK2 as well as the induction of apoptosis in both cell lines by the newly synthesized active agents.

## 2. Results

### 2.1. Chemistry

The acylation of alcohols is one of the most convenient methods to protect their hydroxyl groups [48]. In medicinal chemistry, such simple functionalization of small molecules is often utilized to obtain prodrugs with markedly affected pharmaco-kinetic properties when compared to the parent underivatized active pharmaceutical ingredients (APIs) [49,50,51]. The polar functionality of hydroxyl-containing drugs when masked by acyl moiety derived from, i.e., long-chain fatty acids, became more lipophilic. Moreover, according to the Lipinski’s ‘rule of five’ [52,53,54] lipophilicity is one of the prime physicochemical parameters that exerts great impact on the transport of the organic compounds across the cellular membranes as well as improving metabolic stability corporis. Obviously, more efficient transcellular permeability of xenobiotics also facilitates their bioavailability and determines the overall pharmacological potency of administered prodrugs.

Taking into account the above-mentioned facts, we envisioned that straightforward esterification of known inhibitors of CK2, which were reported by us to be active in non-cell enzyme assay but almost completely inactive in cell-based assay [35], should enhance their physicochemical properties and gain desired anti-cancer activity. The synthetic pathway used to obtain designed esters **3a**–**d** and **6a**–**d** is presented in Figure 2. In the first step, 4,5,6,7-tetrabromo-1*H*-benimidazole (TBBi, **1**) or 4,5,6,7-tetrabromo-2-methyl-1*H*-benzimidazole (2MeTBBi, **4**) were prepared through bromination of corresponding benzoazoles, according to the methods described by us previously [55,56]. Next, *N*-Alkylation of **1** or **4** with 8 equiv of 3-bromopropan-1-ol as the alkylating agent was carried out by the modification of the reported procedures [34,57,58]. Initially, two bases (DBU or K_2_CO_3_) and two aprotic polar solvents (CH_3_CN or DMF) were tested to establish optimal reaction conditions. The highest yields of desired products **2** (up to 73%) and **5** (up to 80%) were obtained when both the reagents were reacted in the presence of 2 equiv of anhydrous K_2_CO_3_ suspended in CH_3_CN and refluxed for four days. In the last step of the synthesis, esters **3a**–**d** and **6a**–**d** were synthesized using standard chemical methods by DMAP-catalyzed acylation of the respective alcohols **2** or **5** with appropriate acyl chloride (butyryl, hexanoyl, octanoyl or laurynoyl chloride) in DCM at 30 °C. Since the desired products **3a**–**d** and **6a**–**d** were obtained in moderate yields (44 to 62%), we decided to change the reaction conditions and try the biocatalytic approach. The implementation of biocatalysts into tailored chemical routes to shorten the number of synthesis steps, enhance efficiency and reduce waste is the standard synthetic strategy utilized in modern organic chemistry. In this context, lipases are the most often used biocatalysts in the esterification of alcohols. This phenomenon mainly stems from their high catalytic activity and thermal stability in organic solvents as well as wide substrates specificity toward xenobiotic substances. It is worth mentioning that in our previous synthetic campaigns the enzymatic method based on lipase catalysis was utilized to obtain optically pure CK2 inhibitors possessing both 4,5,6,7-tetrabromo-1*H*-benzotriazole (TBBt) [59,60] and TBBi [61] scaffolds. In this study, the biocatalytic approach allowed us to obtain final compounds with higher yields (58 to 88%) than when using convenient esterification conditions, while vinyl esters were used as acyl donors, instead of toxic and highly corrosive acyl chlorides. To achieve this goal, the catalytic performance of three different commercially available lipases, including Amano AK from *Pseudomonas fluorescens* (Amano Enzyme Inc., Nagoya, Japan), Amano PS from *Burkholderia cepacia* (Amano Enzyme Inc., Nagoya, Japan) and Novozym 435 (Novozymes A/S, Bagsvaerd, Denmark), an immobilized preparation of thermostable lipase B from the yeast *Candida antarctica* (CAL-B), were screened in toluene, acetonitrile and dichloromethane, respectively. No reaction or low conversions were observed when Amano AK or Amano PS were used as biocatalysts. In contrast, Novozym 435 turned out to catalyze transesterifications of alcohols **2** and **5** in all tested solvents at excellent >99% conversions after 24 h. All synthesized esters **3a**–**d** and **6a**–**d** and novel intermediate **5** were characterized by melting points, high resolution mass spectrometry (HRMS) and spectroscopic techniques (^1^H NMR and ^13^C NMR).

### 2.2. Biological Evaluation

#### 2.2.1. Inhibition of Recombinant CK2 and PIM1 

Inhibition of the human CK2 catalytic subunit (CK2α) and CK2 holoenzyme (CK2 α2β2) by the previously and newly obtained compounds was evaluated using radiometric assay (Table 1). Considering that CK2 inhibitors often demonstrate inhibitory activity towards PIM1 kinase [56], an inhibition of this enzyme by the tested compounds was also evaluated. The synthetic peptide RRRADDSDDDDD was used as the substrate of CK2 and peptide ARKRRRHPSGPPTA as the substrate of PIM1. Evaluation of potential inhibitory activity of the obtained derivatives was performed in the presence of 5 µM and 10 μM of the respective compound and the residual activity of CK2α, CK2 holoenzyme and PIM1 was normalized to the uninhibited protein activity. Furthermore, IC_50_ values were determined for the most active compounds at eight concentrations in the range of 0.001 to 1600 µM. The experimental data were fitted to sigmoidal dose–response (variable slope) Y = Bottom + (Top-Bottom)/(1 + 10^((LogIC_50_ − X)*HillSlope) equation in GraphPad Prism (Prism 9, version 9.0.1, San Diego, CA, USA) and *K*_i_ values were calculated using Cheng and Prusoff equation: *K*_i_ = IC_50_/(1 + [S]/*K*_m_) [62]. Among the newly obtained compounds, derivative **5** is the most efficient inhibitor of CK2 with *K*i values equal 0.19 µM and 0.06 µM for CK2α and CK2 holoenzyme, respectively. 

Additionally, molecular docking of **2** and **5** to human CK2α (PDB: 7A4C [63]) was performed (for details, see Appendix A) and binding modes were proposed. Both inhibitors bind similarly in the active site of CK2α with a similar docking score concerning binding affinity energies expressed as Δ*G*_calc_ (kcal/mol) (–7.0 for **2** and –7.1 for **5**). In both predicted docking models, the halogen-bonding interactions of two bromine atoms with Glu114 and Val116 residues were found, which are characteristic for polybrominated benzimidazoles as CK2 inhibitors. However, compound **5** is slightly twisted so that the methyl group is pointing towards Phe113 and Asp175. In the case of **2**, two polar interactions between the ligands’ hydroxyl group and the side chains of Asp175 (carboxyl group, 2.9 Å) and Ser51 (hydroxyl group, 2.0 Å) were found. For **5**, one polar interaction between the distal hydroxyl group of the ligand molecule and His160 (carbonyl group, 2.8 Å) was detected. The obtained results of enzymatic activity assays revealed that esters of TBBi- and 2MeTBBi-based inhibitors decrease catalytic activity of both CK2 and PIM1 much less efficiently than parental compounds, which can be rationalized by the loss of H-bonding interactions within key amino acid residues of the CK2 active site due to masked hydroxyl moiety. Moreover, there is a clear tendency of the elongation of fatty acid chains in TBBi and 2MeTBBi esters corresponding to a decrease of their inhibitory activity towards CK2 and PIM1 kinases. Probably, it is also the restricted size of the pocket nearby the TBBi binding site that forces these analogs to adopt different conformations which are unable to enter deeply into this cavity due to the bulky nature of acyls and steric clashes within the protein structure. 

#### 2.2.2. Cytotoxic Effect of TBBi Derivatives toward Breast Cancer Cells

We performed our studies on two commercially available human breast cancer cell lines, MCF-7 and MDA-MB-231. Cells were tested for *Mycoplasma hominis contamination* to avoid biased interpretation of the experiments (Appendix A).

To test the anti-cancer effect of the TBBi derivatives, we treated the MCF-7 and MDA-MB-231 cells with the newly synthesized esters and previously reported compounds [34,57] at the concentration range of 3.125 to 200 µM for 24 h, 48 h and 72 h. Additionally, the range of concentrations was extended by additional lower concentrations in MCF-7 for compound **4** with the minimum concentration of 0.195 µM. The representative plots demonstrating sigmoidal dose–response curves for compounds **1**, **2**, **3a**–**c**, **4**, **5** and **6a**–**c** are shown in Figure 3. The EC_50_ values, describing the half maximal effective concentration of each tested compound, were calculated and are summarized in Table 2. Using MTT viability assay, we showed that all tested compounds significantly decreased the viability of both MCF-7 and MDA-MB-231 cells, except for compounds **3d** and **6d** (with EC_50_ > 100 µM). We observed that the EC_50_ values increase in both cell lines with the elongation of fatty acids chain length in the tested compounds, thus strongly correlating with the decrease of their solubility in biological media as characterized by log*S* parameter (Table 2). The second reason for the decreased biological activity of the longer-chain aliphatic esters might be attributed to their greater resistance toward intracellular hydrolysis, which directly influences the slower release of the active form of the corresponding CK2 inhibitors. Interestingly, compounds with the methyl group, i.e., the derivatives from **4** to **6c**, were more cytotoxic towards both cell lines than their counterparts, i.e., compounds from **1** to **3c**. The lowest EC_50_s with the values of 2.66 µM and 4.73 µM were obtained for compound **4** for MCF-7 and MDA-MB-231, respectively, after 72-h treatment. The results confirm our previous data describing 2MeTBBi as a more cytotoxic agent than the parental compound TBBi [34]. The highest EC_50_ values were obtained for all the tested compounds after 24 h of incubation. However, none of the tested compounds, except for compound **5**, decreased the viability of both cell lines in a time-dependent manner (Figure 4).

#### 2.2.3. Intracellular Inhibition of Protein Kinase CK2 in MCF-7 and MDA-MB-231 Cells

To confirm the intracellular inhibition of CK2 in both cell lines by the tested compounds, we evaluated a site-specific phosphorylation of Ser529 and Ser129 in p65 NF-κB (nuclear factor kappa-light-chain-enhancer of activated B-cells) and Akt1, respectively, after 24 h of treatment (Figure 5). We observed an inhibition of CK2 in both cell lines; however, the lower efficiency of CK2 inhibition was detected in MCF-7 considering the extent of Akt1 phosphorylation. The relative level of the phosphorylation of the tested proteins in MDA-MB-231 cells was in the range of 0.32 to 0.80 for p-Ser529 p65 and 0.41 to 0.73 for p-Ser129 Akt1, whereas in MCF-7 cells it was in the range of 0.33 to 0.78 for p-Ser529 p65 and 0.82 to 1.23 for p-Ser129 Akt1, respectively. The lowest phosphorylation levels of the tested proteins were detected in MDA-MB-231 cells for compound **6b** with the relative level of 0.41 for Ser129 Akt1 as well as for compound **6c** with the relative level of 0.32 for p-Ser529 p65. Among the tested proteins, the lowest extent of phosphorylation in MCF-7 cells was detected for **6b** with the relative level of 0.33 for p-Ser529 p65, and for **6a** with the relative level of 0.82 for Ser129 Akt1. 

Considering an inefficient inhibition of CK2 in MCF-7, we evaluated an inhibition of CK2 in this cell line after 72 h. The obtained results showed that all tested compounds affected the activity of CK2 in MCF-7 cells after 72 h of treatment. The relative levels were in the range of 0.17 to 0.8 for p-Ser529 p65 and 0.19 to 0.79 for p-Ser129 Akt1.

The most effective inhibition of CK2 (the relative level of p-Ser529 p65 and p-Ser129 Akt1 equal to 0.17 and 0.19, respectively) was demonstrated for compound **3c** (Figure 6). Interestingly, a derivative **6a** with the same cytotoxic activity as the compound **1** (similar EC_50_s) induced an inhibition of CK2 in MCF-7 more effectively with the relative value of 0.19 and 0.30 for p-Ser529 p65 and p-Ser129 Akt1, respectively. 

#### 2.2.4. Induction of Apoptosis in MCF-7 and MDA-MB-231 Cells

In order to evaluate the proapoptotic properties of the newly synthesized derivatives, we analyzed the level of cleaved PARP1 [c-PARP1, nuclear poly (ADP-ribose) polymerase 1], the hallmark of apoptotic cells [64]. After 24-h treatment, the significantly elevated level of c-PARP1 was detected only in MDA-MB-231 cells with the highest relative level of 7.14 detected in cells treated with compound **6c** (Figure 7). 

Considering the lack of significant effect of the tested compounds in MCF-7 cells after 24-h treatment, we additionally evaluated c-PARP1 level in this cell line after 72 h of treatment. The significant increase of the c-PARP1 level was detected in MCF-7 cells treated with the derivatives **3c** and **6a**–**c** (Figure 8). It should be noted that the highest level of c-PARP up to almost 20-fold higher than in the control was detected in the cell culture treated with a derivative **6b**. Moreover, we noticed an increased level of blebbing in both examined cell lines after 72-h incubation with **3c** and **6a**–**c**, which additionally proved the progression of apoptotic cell death (Appendix A).

Additionally, assaying annexin V-binding to phosphatidylserine by means of flow cytometry confirmed that derivatives **3c** and **6a**–**c** induced apoptosis in both cell lines (Figure 9, Appendix A). The obtained results indicated that compounds **6a**–**c** were more effective in the induction of apoptosis in MCF-7 than **1**, **5** and **3c** after 24-h and 72-h treatment with the highest number of apoptotic cells, i.e., 57.5% after 72-h treatment with derivative **6b** (Figure 9b). These results are in a good agreement with the Western blot analysis for c-PARP1. Moreover, these results strongly correspond to the inhibition of CK2 in the examined cell line since compounds **3c** and **6a**–**c** effectively induced the inhibition of CK2 in MCF-7. It is noteworthy that although compounds **1** and **6a** demonstrated similar EC_50_ values, the mechanism of action of these inhibitors within MCF-7 cells differs. The treatment of the cells with parental compound (**1**, TBBi) resulted in an increase in the level of necrotic cells (25% of total cells) after 72 h, whereas treatment with its octanoate derivative **3c** and compounds **6a**–**c** led to predominantly apoptotic cells (Figure 9a). These data are in a good agreement with the well-established role of CK2 in the anti-apoptotic signalling pathways [24]. Considering that MCF-7 cells demonstrate the deficiency of caspase 3, a key regulator of apoptosis [65], and thus are difficult in entering into apoptosis, the obtained results indicate that the newly obtained derivatives show promising anti-cancer properties. 

Moreover, all the tested compounds also effectively induced apoptosis in MDA-MB-231 cells with the highest level of apoptotic cells after 24 h (32.5%) and 72 h (50.5%) of treatment with compound **3c**. Similar to the results obtained for MCF-7 cells, compound **6b** acted in MDA-MB-231 cells as an effective inducer of apoptosis with the amount of apoptotic cells equal to 31.5% and 49% after 24 h and 72 h of treatment, respectively. 

## 3. Discussion

A series of novel acyl derivatives of 3-(4,5,6,7-tetrabromo-1*H*-benzimidazol-1-yl)propan-1-ols was synthesized by *N*-alkylation of TBBi or 2MeTBBi with 3-bromopropan-1-ol, and subsequent acylation of hydroxyl group employing fatty acids with different aliphatic chain lengths. All esters were obtained with satisfactory yields (44 to 62%) when acyl chlorides were used under convenient DMAP-catalysed conditions. However, the application of lipases as catalysts and vinyl esters as irreversible acyl donors allowed to obtain desired prodrug-like inhibitors in higher yields (58 to 88%) after employing a much simpler work-up procedure, which excluded tedious, time- and labour-consuming liquid−liquid extraction.

The results of in vitro kinetics studies revealed that esters of TBBi- and 2MeTBBi-based inhibitors decrease the catalytic activity of recombinant CK2 and PIM1 isolated enzymes to a significantly lower extent than parental compounds tested previously by our group [35] and others [66]. These findings confirmed that efficient inhibitors of recombinant CK2 should possess (nearby hydrophobic TBBi scaffold) a polar carboxyl or hydroxyl moiety that are prone to form strong hydrogen bonding similarly to the phosphate groups present in ATP physiological substrate. 

We concluded that TBBi derivatives with methyl group at *C*2 atom, i.e., compounds **4**, **5** and **6a**–**c**, decreased the viability of both breast cancer cell lines more efficiently than their counterparts **1**, **2** and **3a**–**c**. Our results also support previous finding that 2MeTBBi is a more potent cytotoxic agent towards MCF-7 and MDA-MB-231 than model TBBi [34]. Moreover, the results presented herein demonstrate that the elongation of the fatty acid chain in ester derivatives of TBBi and 2MeTBBi (above hexanoate moiety) decreases the cytotoxic properties of the modified inhibitors with the highest EC_50_ values obtained for the respective laurinates **3d** and **6d**. This strongly corresponds to the limited solubility of these compounds in biological media (the log*S* values decrease with the elongation of fatty acids) as well as with hypothetical lower rates of enzyme-catalyzed intracellular hydrolysis of more bulky acyl derivatives. In this case, probably longer-chain esters are also less sensitive to low pH conditions and thus more stable in the acidic microenvironment of cancer cells [67]. To prove our assumption as well as to overcome drawbacks resulting from precipitation of the tested compounds in the cell culture media, we decided to study their mechanism of action arbitrarily in concentrations corresponding to their EC_50_ values. This attempt allowed us to compare the results of biological activity of all tested compounds and proved dependency of their lipophilicity on intracellular inhibition of CK2. It was clear that esters characterized by higher log*P* values could penetrate the cell membranes more efficiently, and thus exhibited improved intracellular inhibition of CK2. Consequently, compounds **3c** and **6a**–**c** turned out to prevail over the corresponding alcohols **2** and **5** in terms of CK2 inhibition in both treated breast cancer cell lines. The obtained data support the hypothesis of hydrolysis of **3c** into **2** and **6a**–**c** into **5** in the cells’ microenvironment, considering that both compounds **2** and **5** inhibit CK2α with low *K*_i_ values. The plausible mechanism of action of the most potent prodrug-like 2MeTBBi derivative **6b** as anti-cancer agent is presented in Figure 10. 

Moreover, the results concerning CK2-mediated phosphorylation of Akt and NF-κB strongly correlate with the proapoptotic properties of the best intracellular inhibitors, confirming the anti-apoptotic role of this kinase. It has been shown that CK2 and its substrates, including kinase Akt, protect cells from apoptosis by phosphorylating a wide range of proteins involved in the apoptotic response [68]. Consequently, inhibition of CK2 leads to apoptosis induction in many types of cancer cells. In turn, studies on the activation of NF-κB in mammary glands demonstrated the involvement of CK2 in this process, and it is of note that CK2 is also involved in the phosphorylation of p65 directly, thereby influencing its activity [69]. The indicated differences in the efficacy of novel CK2 inhibitors between MCF-7 and MDA-MB-231 could be related to the caspase-3 deficiency in MCF-7 cells that consequently undergo cell death that lacked typical apoptotic properties [70]. Moreover, it should be also emphasized that the obtained results are promising considering that both studied cell lines are difficult to enter into apoptosis, since the MDA-MB-231 cell line has status p53^−/−^ which makes it less prone to enter apoptosis. This protein is a transcription factor stabilized in the cell in case of DNA damage. It is responsible for stopping the cell cycle in the G1 phase, activation of repair mechanisms and apoptosis of cells with damaged DNA.

## 4. Materials and Methods

### 4.1. Chemistry

#### 4.1.1. General Methods

Commercially available reagents were used as purchased without additional pretreatment and/or purification. Melting points were determined with an MPA100 Optimelt SRS apparatus (Stanford Research Systems, Sunnyvale, CA, USA) and are uncorrected. Nuclear magnetic resonance (NMR) spectra were recorded using a Varian NMR System 500 MHz spectrometer (Varian Inc., Palo Alto, CA, USA). Chemical shifts (*δ*) are reported in parts per million (ppm) related to residual deuterated dimethyl sulfoxide (DMSO-*d*_6_, *δ*H 2.48 ppm and *δ*C 39.51 ppm) as internal standard; signal multiplicities (splitting patterns) are reported using the following abbreviations: s = singlet, d = doublet, t = triplet, quin = quintet, sxt = sextet, m = multiplet. Thin-layer chromatography (TLC) was carried out on aluminum plates with silica gel containing a fluorescent F_254_ indicator (Kieselgel 60 F_254_, Merck, Darmstadt, Germany) (0.2 mm thickness film) and the detection of compounds was performed with short-wave UV light at 254 nm. Silica gel with grain size 40 to 63 μm was used for preparative column chromatography. Fourier transform mass spectrometry (FTMS) was carried out on Q Exactive Hybrid Quadrupole-Orbitrap Mass Spectrometer, ESI (electrospray) with spray voltage 4.00 kV at IBB PAS Warsaw (Waters Corporation, Milford, MA, USA). 

#### 4.1.2. General Procedure for *N*-alkylation of **1** and **4**

To a suspension of the appropriate 4,5,6,7-tetrabromobenzoazole (**1** or **4**, 1.5 mmol) in acetonitrile (20 mL), anhydrous K_2_CO_3_ (414 mg, 3.0 mmol, 2.0 eq.) and 3-bromopropan-1-ol (0.72 mL, 8.0 eq.) were added. The reaction mixture was stirred at reflux for 96 h. The progress of the reaction was monitored by TLC [CHCl_3_/MeOH (95:5, *v*/*v*)]. The crude mixture was concentrated under reduced pressure and the remaining residue was diluted with CHCl_3_ (50 mL) and washed with H_2_O (3 × 30 mL). The organic layer was dried over anhydrous MgSO_4_, filtered, and the solvent was removed under reduced pressure. The crude product was purified by column chromatography on silica gel using a gradient of DCM/MeOH (100:0, 95:5, *v*/*v*) mixture as an eluent to afford desired alcohols **2** or **5**. 


*(4,5,6,7-Tetrabromo-1H-benzimidazol-1-yl)propan-1-ol (**2**)*


Product was obtained as a white solid, 73% yield.

m.p. 196 to 198 °C {lit. 196.4 to 197.2 °C, [58]}; MS [M+H]^+^ m/z calculated for C_10_H_9_Br_4_N_2_O^+^ 492.7402, found 492.6213.


*(4,5,6,7-Tetrabromo-2-methyl-1H-benzimidazol-1-yl)propan-1-ol (**5**)*


Product was obtained as a white solid, 80% yield.

m.p. 227 to 230 °C; HRMS [M+H]^+^ m/z calculated for C_11_H_11_Br_4_N_2_O^+^ 506.75585, found 506.75607; ^1^H NMR (500 MHz, DMSO-*d*_6_) δ ppm: 1.83 to 1.90 (m, 2H), 2.58 (s, 3H), 3.41 to 3.47 (m, 2H), 4.44 (t, *J* = 7.6 Hz, 2H), 4.67 (t, *J* = 4.9 Hz, 1H); ^13^C NMR (125 MHz, DMSO-*d*_6_) δ ppm: 14.1, 34.0, 42.5, 57.4, 105.8, 115.2, 119.8, 121.5, 132.1, 142.7, 156.8.

#### 4.1.3. General Procedure for Esterification of **2** and **5**

*Method A (Chemical)*: To a mixture of the appropriate alcohol (**2** for esters **3a**–**d**, and **5** for esters **6a**–**d**, 0.30 mmol) and *N*,*N*-dimethylpyridin-4-amine (DMAP, 0.15 mmol, 0.5 eq.) in DCM (10 mL), the appropriate acyl chloride (0.60 mmol, 2.0 eq.) was added in one portion. The reaction mixture was stirred at 30 °C for 24 h until the substrate was completely consumed according to TLC analysis [CHCl_3_/MeOH (98:2, *v*/*v*)]. The crude mixture was washed with a saturated aqueous solution of NaHCO_3_ (2 × 20 mL), whereas the combined water layers were back-extracted with DCM (2 × 15 mL). The combined organic layers were dried over anhydrous MgSO_4_, filtered, and the solvent was removed under reduced pressure. The crude products were purified by column chromatography on silica gel using a gradient of CHCl_3_/MeOH (100:0, 99:1, *v*/*v*) mixture as an eluent to afford the desired esters **3a**–**d** and **6a**–**d**.

*Method B (Enzymatic)*: To a suspension of the appropriate alcohol (**2** for esters **3b**, **3d**, and **5** for esters **6b**, **6d**, 0.2 mmol) in DCM (12 mL), the appropriate vinyl ester (0.6 mmol, 2.0 eq.) and Novozym 435 (10% *w*/*w*) were added. The reaction mixture was stirred at 30 °C for 24 h. The progress of the reaction was monitored by TLC [CHCl_3_/MeOH (98:2, *v*/*v*)]. The enzyme was filtered off and washed with DCM (3 × 5 mL). Filtrate was washed with H_2_O (3 × 10 mL), the organic layer was dried over anhydrous MgSO_4_, filtered, and the solvent was removed under reduced pressure. The crude products were purified by column chromatography on silica gel using a gradient of CHCl_3_/MeOH (100:0, 99:1, *v*/*v*) mixture as an eluent to afford desired esters **3b**, **3d**, **6b** and **6d**.


*(4,5,6,7-Tetrabromo-1H-benzimidazol-1-yl)propyl butanoate (**3a**)*


Procedure A: product was obtained as a white solid, 54% yield.

m.p. 96 to 97 °C; HRMS [M+H]^+^ m/z calculated for C_14_H_15_Br_4_N_2_O_2_^+^ 562.78206, found 562.78263; ^1^H NMR (500 MHz, DMSO-*d*_6_) *δ* ppm: 0.79 (t, *J* = 7.3 Hz, 3H), 1.39 (sxt, *J* = 7.3 Hz, 2H), 2.06 (t, *J* = 7.3 Hz, 2H), 2.10 to 2.17 (m, 2H), 4.03 (t, *J* = 5.9 Hz, 2H), 4.57 (t, *J* = 6.9 Hz, 2H), 8.47 (s, 1H); ^13^C NMR (125 MHz, DMSO-*d*_6_) δ ppm: 13.4, 17.7, 30.3, 35.1, 44.0, 61.1, 106.5, 116.6, 120.4, 122.4, 131.4, 143.7, 149.0, 172.5.


*3-(4,5,6,7-Tetrabromo-1H-benzimidazol-1-yl)propyl hexanoate (**3b**)*


Procedure A: product was obtained as a white solid, 58% yield.

Procedure B: product was obtained as a white solid, 67% yield. 

m.p. 65 to 67 °C; HRMS [M+H]^+^ m/z calculated for C_16_H_19_Br_4_N_2_O_2_^+^ 590.81336, found 590.81400; ^1^H NMR (500 MHz, DMSO-*d*_6_) *δ* ppm: 0.81 (t, *J* = 7.3 Hz, 3H), 1.06 to 1.14 (m, 2H), 1.15 to 1.23 (m, 2H), 1.32 (quin, *J* = 7.3 Hz, 2H), 2.02 (t, *J* = 7.5 Hz, 2H), 2.09 to 2.17 (m, 2H), 4.04 (t, *J* = 5.9 Hz, 2H), 4.58 (t, *J* = 6.8 Hz, 2H), 8.47 (s, 1H); ^13^C NMR (125 MHz, DMSO-*d*_6_) *δ* ppm: 13.8, 21.7, 23.9, 30.2, 30.6, 33.2, 44.1, 61.2, 106.4, 116.6, 120.4, 122.3, 131.4, 143.7, 148.9, 172.6.


*3-(4,5,6,7-Tetrabromo-1H-benzimidazol-1-yl)propyl octanoate (**3c**)*


Procedure A: product was obtained as a white solid, 58% yield.

m.p. 48 to 50 °C; HRMS [M+H]^+^ m/z calculated for C_18_H_23_Br_4_N_2_O_2_^+^ 618.84466, found 618.84567; ^1^H NMR (500 MHz, DMSO-*d*_6_) *δ* ppm: 0.82 (t, *J* = 7.1 Hz, 3H), 1.07 to 1.26 (m, 8H), 1.27 to 1.34 (m, 2H), 2.03 (t, *J* = 7.6 Hz, 2H), 2.09 to 2.17 (m, 2H), 4.04 (t, *J* = 5.9 Hz, 2H), 4.04 (t, *J* = 5.9 Hz, 2H), 4.58 (t, *J* = 6.8 Hz, 2H), 8.48 (s, 1H); ^13^C NMR (125 MHz, DMSO-*d*_6_) *δ* ppm: 13.9, 22.0, 24.2, 28.3, 28.4, 30.2, 31.1, 33.3, 44.2, 61.2, 106.5, 116.6, 120.4, 122.3, 131.4, 143.7, 148.9, 172.6.


*3-(4,5,6,7-Tetrabromo-1H-benzimidazol-1-yl)propyl dodecanoate (**3d**)*


Procedure A: product was obtained as a white solid, 47% yield.

Procedure B: product was obtained as a white solid, 58% yield.

m.p. 59 to 60 °C; HRMS [M+H]^+^ m/z calculated for C_22_H_31_Br_4_N_2_O_2_^+^ 674.90726, found 674.90738; ^1^H NMR (500 MHz, DMSO-*d*_6_) *δ* ppm: 0.81 (t, *J* = 6.8 Hz, 3H), 1.11 to 1.25 (m, 16H), 1.27 to 1.36 (m, 2H), 2.03 (t, *J* = 7.3 Hz, 2H), 2.10 to 2.18 (m, 2H), 4.04 (t, *J* = 5.9 Hz, 2H), 4.58 (t, *J* = 6.6 Hz, 2H), 8.47 (s, 1H); ^13^C NMR (125 MHz, DMSO-*d*_6_) δ ppm: 13.9, 22.1, 24.2, 28.4, 28.6, 28.7, 28.8, 28.9, 29.0, 30.2, 31.3, 33.3, 44.1, 61.2, 106.4, 116.6, 120.4, 122.3, 131.4, 143.7, 149.0, 172.6.


*3-(4,5,6,7-Tetrabromo-2-methyl-1H-benzimidazol-1-yl)propyl butanoate (**6a**)*


Procedure A: product was obtained as a white solid, 62% yield.

m.p. 81 to 82 °C; HRMS [M+H]^+^ m/z calculated for C_15_H_17_Br_4_N_2_O_2_^+^ 576.79771, found 576.79838; ^1^H NMR (500 MHz, DMSO-*d*_6_) *δ* ppm: 0.83 (t, *J* = 7.3 Hz, 2H), 1.45 (sxt, *J* = 7.3 Hz, 2H), 2.02 to 2.09 (m, 2H), 2.15 (t, *J* = 7.3 Hz), 2.57 (s, 3H), 4.10 (t, *J* = 5.9 Hz, 2H), 4.48 (t, *J* = 7.6 Hz, 2H); ^13^C NMR (125 MHz, DMSO-*d*_6_) δ ppm: 13.4, 14.1, 17.8, 29.8, 35.2, 42.4, 61.1, 105.7, 115.3, 119.9, 121.6, 132.1, 142.7, 156.6, 172.6.


*(4,5,6,7-Tetrabromo-2-methyl-1H-benzimidazol-1-yl)propyl hexanoate (**6b**)*


Procedure A: product was obtained as a colourless oil which slowly solidified, 44% yield.

Procedure B: product was obtained as a colourless oil which slowly solidified, 84% yield.

m.p. 47 to 50 °C; HRMS [M+H]^+^ m/z calculated for C_17_H_21_Br_4_N_2_O_2_^+^ 604.82901, found 604.82967; ^1^H NMR (500 MHz, DMSO-*d*_6_) *δ* ppm: 0.83 (t, *J* = 7.3 Hz, 3H), 1.14 to 1.20 (m, 2H), 1.20 to 1.26 (m, 2H), 1.39 (quin, *J* = 7.3 Hz, 2H), 2.03 to 2.10 (m, 2H), 2.11 to 2.16 (m, 2H), 2.57 (s, 3H), 4.10 (t, *J* = 5.8 Hz, 2H), 4.50 (t, *J* = 7.4 Hz); ^13^C NMR (125 MHz, DMSO-*d*_6_) δ ppm: 13.8, 14.1, 21.8, 24.0, 29.8, 30.6, 33.3, 42.5, 61.2, 105.8, 115.3, 120.0, 121.6, 132.1, 142.7, 156.7, 172.7.


*3-(4,5,6,7-Tetrabromo-2-methyl-1H-benzimidazol-1-yl)propyl octanoate (**6c**)*


Procedure A: product was obtained as a white solid, 58% yield.

m.p. 61 to 65 °C; HRMS [M+H]^+^ m/z calculated for C_19_H_25_Br_4_N_2_O_2_^+^ 632.86031, found 632.86095; ^1^H NMR (500 MHz, DMSO-*d*_6_) *δ* ppm: 0.82 (t, *J* = 7.1 Hz, 3H), 1.14 to 1.25 (m, 8H), 1.35 to 1.42 (m, 2H), 2.02 to 2.09 (m, 2H), 2.12 (t, *J* = 7.4 Hz, 2H), 2.57 (s, 3H), 4.10 (t, *J* = 5.9 Hz, 2H), 4.49 (t, *J* = 7.3 Hz, 2H); ^13^C NMR (125 MHz, DMSO-*d*_6_) δ ppm: 13.9, 14.1, 22.0, 24.3, 28.3, 28.4, 29.8, 31.1, 33.3, 42.4, 61.2, 105.7, 115.3, 120.0, 121.5, 132.1, 142.7, 156.6, 172.7.


*(4,5,6,7-Tetrabromo-2-methyl-1H-benzimidazol-1-yl)propyl dodecanoate (**6d**)*


Procedure A: product was obtained as a white solid, 52% yield.

Procedure B: product was obtained as a white solid, 88% yield.

m.p. 69 to 70 °C; HRMS [M+H]^+^ m/z calculated for C_23_H_33_Br_4_N_2_O_2_^+^ 688.92291, found 688.92277; ^1^H NMR (500 MHz, DMSO-*d*_6_) *δ* ppm: 0.81 (t, *J* = 6.8 Hz, 3H), 1.13 to 1.25 (m, 16H), 1.34 to 1.43 (m, 2H), 2.02 to 2.10 (m, 2H), 2.13 (t, *J* = 7.6 Hz, 2H), 2.57 (s, 3H), 4.10 (t, *J* = 5.9 Hz, 2H), 4.49 (t, *J* = 7.6 Hz, 2H); ^13^C NMR (125 MHz, DMSO-*d*_6_) *δ* ppm: 13.9, 14.1, 22.1, 24.3, 28.4, 28.6, 28.7, 28.8, 28.9, 29.0, 29.8, 31.3, 33.3, 42.4, 61.2, 105.7, 115.3, 120.0, 121.5, 132.1, 142.7, 156.6, 172.7.

### 4.2. Biological Evaluation

#### 4.2.1. Reagents and Antibodies

Dimethyl sulphoxide (DMSO), Molecular Biology grade, used as a solvent for all stocks of the chemical agents, was obtained from Roth (Karlsruhe, Germany). All reagents used in flow cytometry analysis were purchased from BD Biosciences Pharmingen (San Diego, CA, USA). The following primary antibodies were used: anti-GAPDH (Millipore, St. Louis, MO, USA; #MAB374, 1:20,000, 30 min, RT (room temperature), anti-p-p65 (Ser529) (Biorbyt Cambridge, UK); #orb14916, 1:500, overnight, +4 °C), anti-p65 (CST, Beverly, MA, USA; #D14E12, 1:1000, overnight, +4 °C), anti-p-Akt1 (Ser129) (Sigma-Aldrich, (St. Louis, MO, USA; #SAB4301414, 1:500, overnight, +4 °C), anti-Akt1 (BD Transduction Lab, San Diego, CA, USA; #610876, 1:500, overnight, +4 °C), and anti-c-RARP1 (CST, Danvers, MA, USA; #9541, 1:750, overnight, +4 °C). The following secondary antibodies were used: HRP-conjugated anti-mouse (Dako, Santa Clara, CA, USA); #P0447, 1:1000, 1 h, RT; Millipore, St. Louis, MO, USA; #AP308P, 1:10,000, 1 h, RT) and anti-rabbit IgG (Dako, Santa Clara, CA, USA); #P0448, 1:2000, 1 h, RT; Millipore, St. Louis, MO, USA; #AP307P, 1:10,000, 1 h, RT). Protease inhibitors were from Roche Applied Science (#11836153001, Mannheim, Germany). Nitrocellulose membrane was from GE Healthcare Life Sciences (Freiburg, Germany) and solvents for HRP reaction (Western Bright Peroxide and Western Bright Quantum) were purchased from Advansta (Menlo Park, CA, USA). ECL reagent was from BioRad (Hercules, CA, USA) and Millipore (St. Louis, MO, USA). Other solvents, reagents and chemicals were purchased from POCH (Avantor Performance Materials, Gliwice, Poland), Merck and Sigma-Aldrich Chemical Company (St. Louis, MO, USA). 

#### 4.2.2. Inhibition of Recombinant CK2 and PIM1

The new TBBi derivatives were tested for their inhibitory activity toward human CK2α, human CK2 holoenzyme and PIM1 using P81 filter isotopic assay as it was described previously [35].

#### 4.2.3. Cell Culture and Agents Treatment

Human Caucasian breast adenocarcinoma cell line, MDA-MB-231 was purchased from ECACC (European Collection of Authenticated Cell Cultures, 92020424)), whereas MCF-7 was purchased from ATCC (American Type Culture Collection, HTB-22)). Jurkat leukaemic T-cell line was kindly provided by Prof. Krzysztof Zabłocki from the Nencki Institute. 

MCF-7 and MDA-MB-231 were cultured in high-glucose Dulbecco’s modified Eagle medium (DMEM, Lonza, Basel, Switzerland) supplemented with 10% fetal bovine serum (FBS, EuroClone, Siziano, Italy)), 2 mM L-glutamine, and antibiotics (100 U/mL penicillin, 100 µg/mL streptomycin) (Sigma-Aldrich Chemical Company (St. Louis, MO, USA). Cells were grown in 75-cm^2^ cell culture flasks (Sarstedt, Nümbrecht, Germany) in a humidified atmosphere of CO_2_/air (5%/95%) at 37 °C. All the experiments were performed in exponentially growing cultures. Stock solutions of the test compounds were prepared in DMSO and stored in −80 °C for a maximum of one month. For the cytotoxicity studies, stock solutions of the tested compounds were diluted 200-fold with the proper culture medium to obtain the final concentrations. 

Jurkat cells were cultured in the RPMI-1640 medium (Gibco, Waltham, MA, USA) with 10% FBS (Gibco, Waltham, MA, USA) and 1% penicillin/streptomycin (Gibco, Waltham, MA, USA) at 37 °C in a humidified atmosphere with 5% CO_2_.

#### 4.2.4. 3-(4,5-Dimethylthiazol-2-yl)-2,5-diphenyltetrazolium Bromide (MTT)-Based Viability Assay

After incubation with the tested compounds, an MTT test was performed as described previously [35]. Optical densities were measured at 570 nm using BioTek microplate reader (Winooski, VT, USA). All measurements were carried out in a minimum of three biological replicates.

#### 4.2.5. Western Blotting

The MCF-7 cells growing exponentially were seeded at 4.8 × 10^5^ cells in 6-cm-diameter plates. Subsequently, compounds were added in a final concentration of 0.5% DMSO. After incubation, floating cells were collected and cell monolayer was washed three times in ice-cold PBS and the cells were scraped, combined with floating cells and lysed in RIPA buffer consisting of 50 mM Tris-HCl (pH 7.4), 1% NP-40, 0.5% sodium deoxycholate, 0.1% SDS, 150 mM NaCl, 2 mM EDTA, 50 mM NaF, 0.2 mM sodium orthovanadate, and protease inhibitors cocktail. Subsequently, cells were sonicated (three times for 15 s), centrifuged at 20,000× *g* for 15 min at 4 °C, and the supernatants were collected and stored at −80 °C. The protein concentration was determined using Bradford assay. Then, supernatants were incubated with the Laemmli buffer for 5 min at 98 °C. Equal amounts of total protein (10 to 20 μg) were analyzed by SDS-PAGE, and subsequently, Western blotting was performed using primary antibodies in blocking buffer containing either 3% BSA (bovine serum albumin) or 3% milk in TBS-T (Tris-buffered saline (10 mM Tris-HCl (pH 8), 150 mM NaCl) containing 0.1% Tween 20 or 0.2% Triton X-100) for 1 h. After overnight incubation at 4 °C with the primary antibodies, the membranes were washed with TBS-T and probed with the respective secondary antibodies. The ECL substrate was used for detection and immunoblots were scanned using G Box Chemi (Syngene, Cambridge, UK) with GeneSys software (Syngene, Cambridge, UK) or were developed with the use of X-rays films. UV-irradiated Jurkat cells were used as a positive control in Western blot analysis of c-PARP1. After 2 h, one-minute UV-irradiated Jurkat leukemic T-cell line was lysed and subjected to analysis.

#### 4.2.6. Densitometry

For densitometry, immunoblots were quantified using the Fiji distribution of the ImageJ 1.52a software (National Institutes of Health, Bethesda, MD, USA, and the University of Wisconsin, Madison, WI, USA). Phosphorylated protein densities were normalized to GAPDH densities, assuming 1 for untreated cells and then they were converted to a percent of the appropriate control.

#### 4.2.7. Detection of Apoptosis by Flow Cytometry

MCF-7 cells were seeded in six-well plates at 1.5 × 10^5^ cells/well. After 24 h of incubation, cells were treated with the tested compounds used in concentrations corresponding to their EC_50_ values. Then, the plates were incubated for 24 h or 72 h. After exposure to the examined compounds, the floating cells were collected, and monolayer cell culture was trypsinized. The cells were centrifugated at 200× *g* at 4 °C for 5 min, washed twice in cold phosphate-buffered saline (PBS), and subsequently suspended in binding buffer at 1 × 10^6^ cells/mL. Subsequently, 100-μL aliquots of the cell suspension were labeled according to the kit manufacturer’s instructions. Briefly, annexin V-fluorescein isothiocyanate and propidium iodide (BD Biosciences, Pharmingen, San Diego, CA, USA) were added to the cell suspension, and the mixture was vortexed and then incubated for 15 min at RT in the dark. A cold binding buffer (900 μL) was then added and the cells were vortexed again and kept on ice. Flow cytometric measurements were performed within 1 h after labeling. Viable, necrotic, early, and late apoptotic cells were detected by flow cytometry using CyFlow Cube 8 (Sysmex, Norderstedt, Germany) flow cytometer.

#### 4.2.8. Cell Cycle Progression by Flow Cytometry

MCF-7 cells were cultured in six-well plates and treated with the tested compounds used in concentrations corresponding to their EC50 values (72 h). After exposure to the compounds, the cells were trypsinized, collected and washed with cold PBS and fixed at −20 °C in 70% ethanol for at least 24 h. Subsequently, the cells were washed in PBS and stained with 50 μg/mL PI (propidium iodide) and 100 μg/mL RNase solution in PBS supplemented with 0.1% *v*/*v* Triton X-100 for 30 min in the dark at the RT. Cellular DNA content was determined by flow cytometry employing FACSCanto II flow cytometer (BD Biosciences, San Jose, CA, USA) flow cytometer. The DNA histograms obtained were analyzed using MacCycle software (Phoenix Flow Systems, San Diego, CA, USA) for evaluation of the distribution of the cells in different phases of the cell cycle.

#### 4.2.9. Statistical Evaluation

Results are represented as mean ± s.e.m. of at least three independent experiments. Statistical analysis was performed using the GraphPad Prism 5.0 software (GraphPad Software Inc., San Diego, CA, USA). Significance was determined using a one-way ANOVA analysis. The statistical significance of differences was indicated in figures by asterisks as follows: * *p* ≤ 0.05, ** *p* ≤ 0.01 and *** *p* ≤ 0.001.

## 5. Conclusions

In summary, considering that many efficient protein kinase CK2 inhibitors derived from tetrabromo-benzimidazole and tetrabromo-benzotriazole exhibit low cytotoxicity, in this study, we aimed at designing of acyl-prodrugs of well-established CK2 inhibitors as a novel therapeutic strategy against breast cancer. Cellular assays on two various breast cancer cell lines rendered promising results for 4-(4,5,6,7-tetrabromo-2-methyl-1*H*-benzimidazol-1-yl)propyl hexanoate (**6b**). This prodrug inhibitor, most probably releasing in cellulo active agent **5**, presented the highest cytotoxic activity, CK2 inhibition in cancer cells, and the best proapoptotic capability, thus being the most promising drug candidate for further in vivo studies. The obtained results also support the idea that CK2 kinase is vital factor for breast cancer cell survival and an appealing molecular drug target useful in the development of novel antineoplastic agents. The main lesson to be drawn from data presented in this paper, and obtained by both in vitro inhibitory evaluations with isolated enzymes and the respective cell viability/proliferation assays, is that in order to preserve high CK2 inhibitory activity in cellulo, and thus achieve desired anti-cancer activity of the TBBi derivatives, the application of a relevant targeted prodrug strategy to mask redundant polar functionality is highly recommended. In this case, the designed TBBi hexanoate prodrug **6b** can release the inhibitor **5** rapidly under intracellular conditions, showing excellent in vitro antitumor activity for achieving a specific and efficient pharmacological effect in targeted breast cancer therapy. 

## Figures and Tables

**Figure 1 ijms-22-06261-f001:**
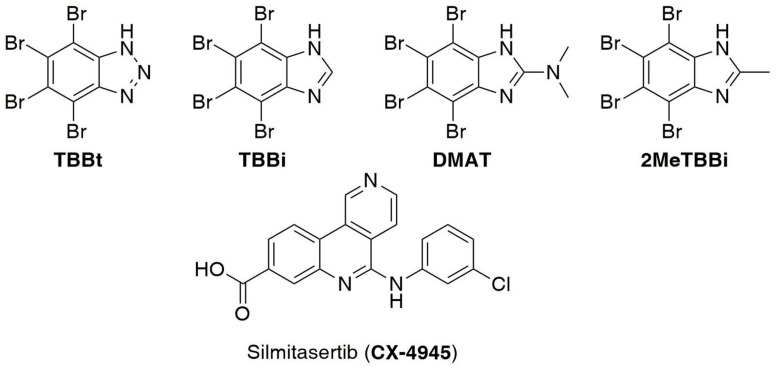
Representative examples of CK2 inhibitors: 4,5,6,7-tetrabromo-1*H*-benzotriazole (TBBt), 4,5,6,7-tetrabromo-1*H*-benzimidazole (TBBi), 4,5,6,7-tetrabromo-*N*,*N*-dimethyl-1*H*-benzimidazol-2-amine (DMAT), 4,5,6,7-tetrabromo-2-methyl-1*H*-benzimidazole (2MeTBBi), and silmitasertib (CX-4945).

**Figure 2 ijms-22-06261-f002:**
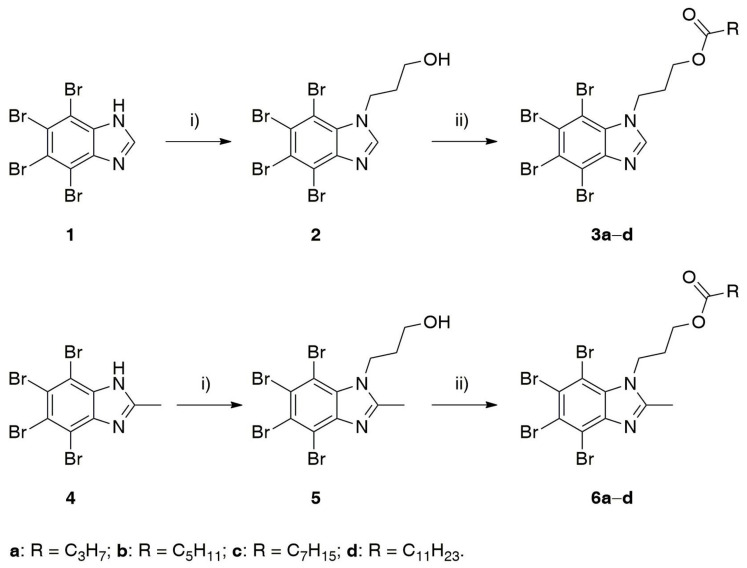
Synthesis of the 3-(4,5,6,7-tetrabromo-1*H*-benzimidazol-1-yl)propyl esters (**3a**–**d**) and 3-(4,5,6,7-tetrabromo-2-methyl-1*H*-benzimidazol-1-yl)propyl esters (**6a**–**d**): (i) 3-bromopropan-1-ol (8 equiv), K_2_CO_3_ (2 equiv), CH_3_CN, reflux, 96 h; (ii) acyl chloride (RCOCl, 2 equiv), DMAP (0.5 equiv), DCM, 30 °C, 24 h or vinyl ester [RC(O)OCH=CH_2_, 2 equiv], Novozym 435 (10% *w*/*w*), DCM, 30 °C, 24 h.

**Figure 3 ijms-22-06261-f003:**
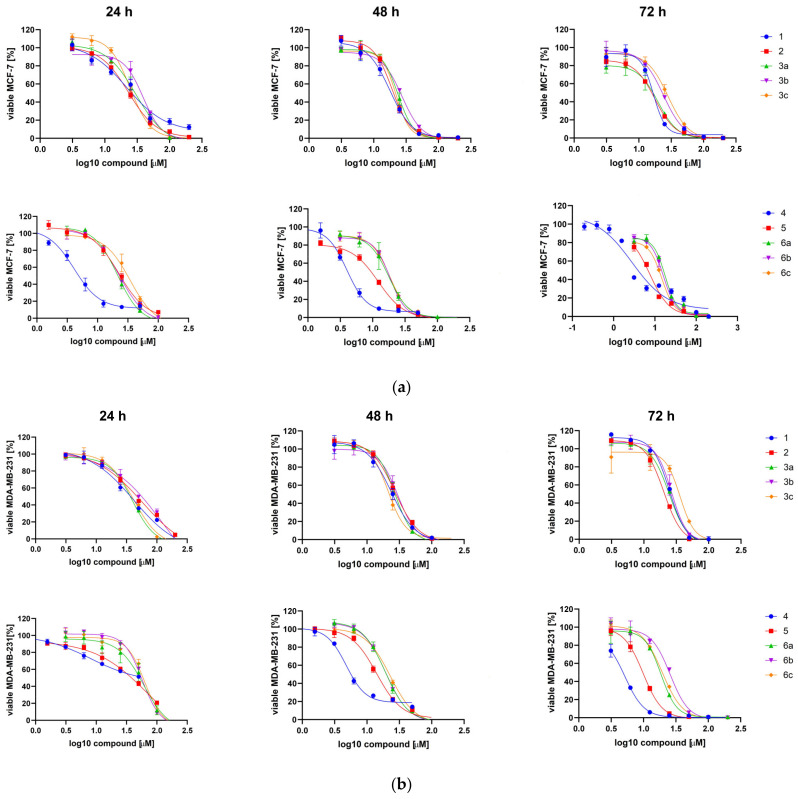
Sigmoidal dose–response curves for compounds **1**, **2**, **3a**–**c**, **4**, **5** and **6a**–**c** determined for MCF-7 (**a**) and MDA-MB-231 (**b**) after 24 h, 48 h and 72 h of treatment. Plots were generated by GraphPad Prism after fitting the MTT data to sigmoidal dose response equation Y = 100/(1 + 10^((LogIC_50_ − X) * HillSlope)).

**Figure 4 ijms-22-06261-f004:**
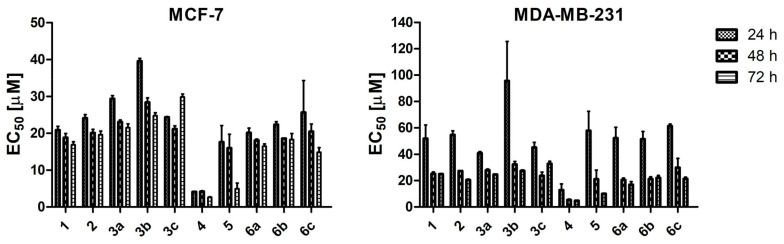
The EC_50_ representing half maximal effective concentration of compounds **1**, **2**, **3a**–**c**, **4**, **5** and **6a**–**c** determined for MCF-7 and MDA-MB-231 after 24, 48 and 72 h of treatment. Graphs represent mean values ± s.e.m.

**Figure 5 ijms-22-06261-f005:**
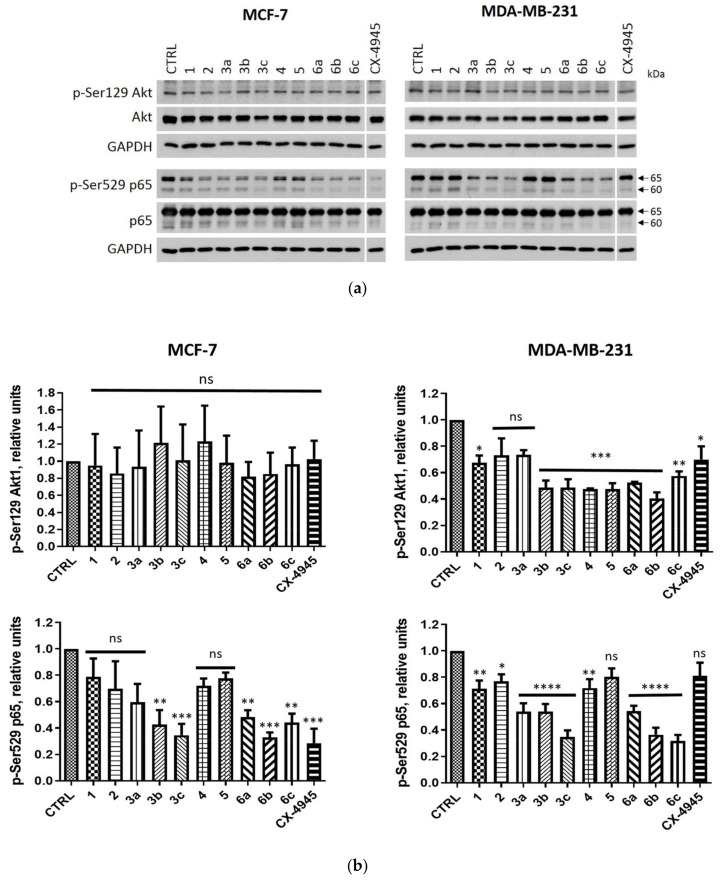
The effects of compounds **1**, **2**, **3a**–**c**, **4**, **5** and **6a**–**c** on the levels of p65 and Akt1 and their phosphorylated forms (p-Ser529 p65 and p-Ser129 Akt1, respectively) in MCF-7 and MDA-MB-231 cell lines after 24-h treatment. (**a**) Western blot analysis of the following proteins in the extracts obtained from MCF-7 and MDA-MB-231 cells treated with the tested compounds used in the concentrations corresponding to their EC_50_ values for 72-h treatment. GAPDH was used as a protein loading control for each sample. (**b**) Densitometry analysis for p-Ser529 p65 and p-Ser129 Akt1 were calculated relative to untreated control (CTRL) cells. CX-4945 was used as a positive control. Graphs represent mean values ± s.e.m. * *p* < 0.05, ** *p* < 0.01, *** *p* < 0.001, **** *p* < 0.0001 relative to control; ns—not significant.

**Figure 6 ijms-22-06261-f006:**
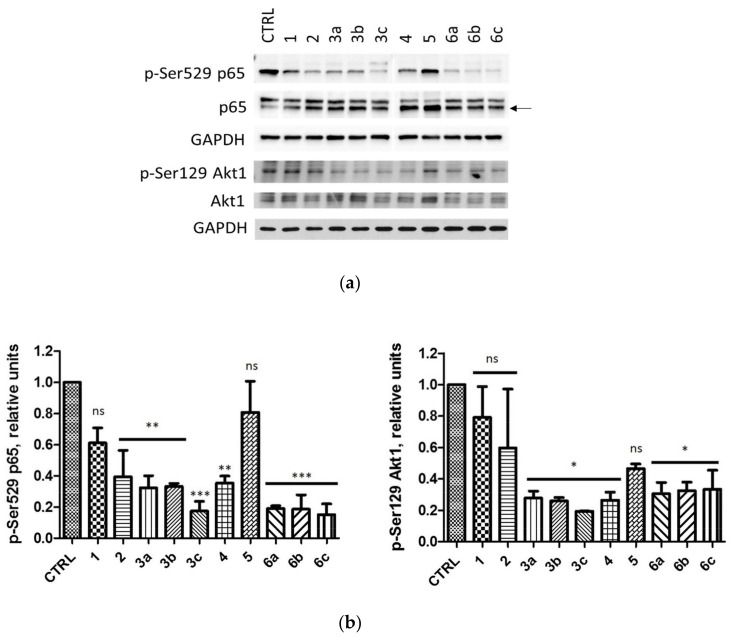
Effects of compounds **1**, **2**, **3a**–**c**, **4**, **5** and **6a**–**c** on the levels of p65 and Akt1 and their phosphorylated forms (p-Ser529 p65 and p-Ser129 Akt1, respectively) in MCF-7 cell line after 72-h treatment. (**a**) Western blot analysis of the following proteins in the extracts obtained from MCF-7 cells after 72-h treatment with the tested compounds used in the concentrations corresponding to their EC_50_ values. GAPDH was used as a protein loading control for each sample. (**b**) Densitometry analysis for p-Ser529 p65 and p-Ser129 Akt1 was performed relative to untreated control (CTRL) cells. Graphs represent mean values ± s.e.m. * *p* < 0.05, ** *p* < 0.01, *** *p* < 0.001 relative to control; ns—not significant.

**Figure 7 ijms-22-06261-f007:**
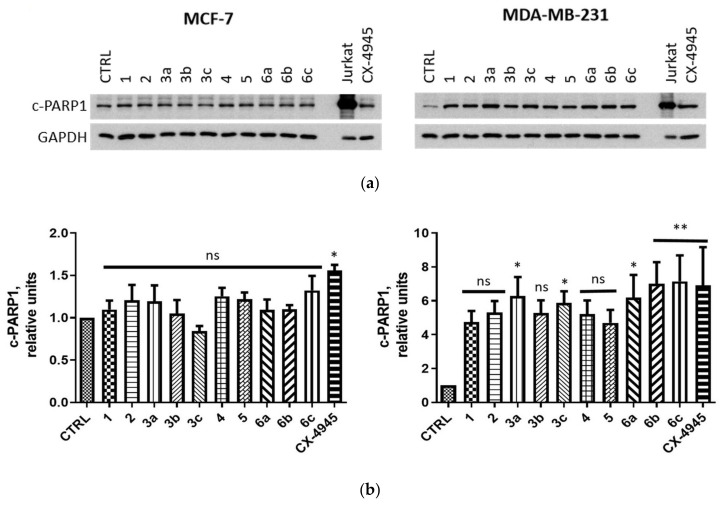
Effect of compounds **1**, **2**, **3a**–**c**, **4**, **5** and **6a**–**c** on the level of cleaved PARP1 (c-PARP1) in MCF-7 and MDA-MB-231 cells. Cells were treated with compounds **1**, **2**, **3a**–**c**, **4**, **5** and **6a**–**c** used in the concentrations corresponding to their EC_50_ values for 24 h and subsequently the respective crude extracts were subjected to Western blot analysis (see “Western Blotting”) with the specific antibodies against c-PARP. GAPDH was used as a protein loading control for each sample. (**a**) The representative blots. UV-irradiated Jurkat cells were used as a positive control (see “Western Blotting”). (**b**) Densitometry analysis for c-PARP1. Untreated cells served as the reference point (one relative unit). Graphs represent mean values ± s.e.m. * *p* < 0.05, ** *p* < 0.01 relative to control; ns—not significant.

**Figure 8 ijms-22-06261-f008:**
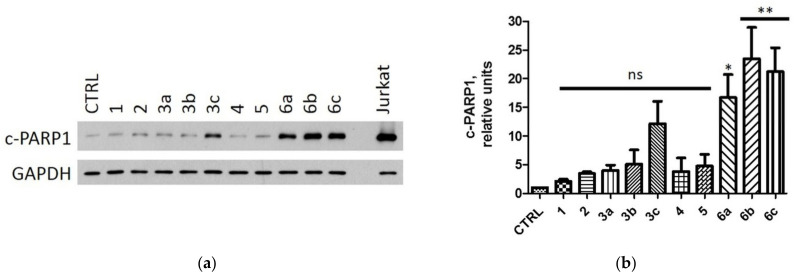
Effect of compounds **1**, **2**, **3a**–**c**, **4**, **5** and **6a**–**c** on the level of cleaved PARP1 (c-PARP1) in MCF-7 cells. MCF-7 cells were treated with compounds **1**, **2**, **3a**–**c**, **4**, **5** and **6a**–**c** used in the concentrations corresponding to their EC_50_ values for 72 h, and subsequently, the respective extracts were subjected to Western blot analysis (see “Western Blotting”). GAPDH was used as a protein loading control for each sample. (**a**) The representative blots. UV-irradiated Jurkat cells were used as a positive control (see “Western Blotting”). (**b**) Densitometry analysis for c-PARP1. Untreated cells served as the reference point (one relative unit). Graphs represent mean values ± s.e.m. * *p* < 0.05, ** *p* < 0.01 relative to control; ns—not significant.

**Figure 9 ijms-22-06261-f009:**
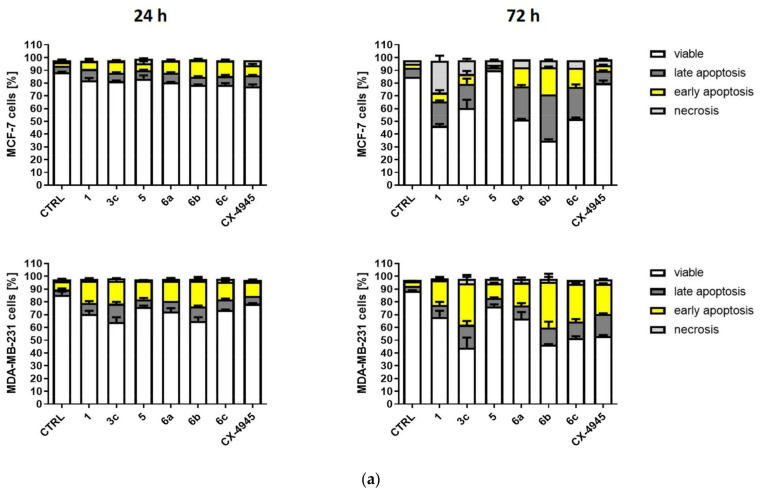
Effect of compounds **1**, **3c**, **5** and **6a**–**c** on progression of apoptosis/necrosis in MCF-7 and MDA-MB-231 cells. (**a**) Percentage of viable, early and late apoptotic, and necrotic cells, after 24 h and 72 h of treatment with the tested compounds used in the concentrations corresponding to their EC_50_ values for 72-h treatment. (**b**) Percentage of apoptotic cells (sum of the percentage of cells in early and late apoptosis) after 24-h and 72-h treatment with the tested compounds. Cells were stained with annexin V-FITC and PI (propidium iodide). CX-4945 was used as a positive control. The data were determined by CyFlow Cube 8, (Sysmex, Norderstedt, Germany) flow cytometer. Graphs represent mean values ± s.e.m. * *p* < 0.05, ** *p* < 0.01, *** *p* < 0.001 relative to control; ns—not significant.

**Figure 10 ijms-22-06261-f010:**
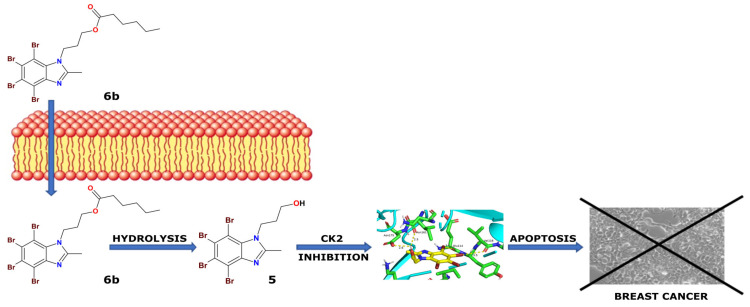
Hypothetical mechanism of action of the designed prodrug-like ester **6b** on breast cancer cells that, after entering the cell, is hydrolyzed to the active form of CK2 inhibitor **5**.

**Table 1 ijms-22-06261-t001:** Inhibition of CK2 and PIM1 by compounds **1**, **2**, **3a**–**c** and **4**, **5**, **6a**–**c**.

Cpd.	Residual Activity of the Enzyme [%]						*K*_i_ [µM]		
	CK2α		CK2 α_2_β_2_		PIM1		CK2α	CK2 α_2_β_2_	PIM1
	5 µM	10 µM	5 µM	10 µM	5 µM	10 µM			
**1**	19.3 ± 0.2	12.4 ± 0.1	9.6 ± 2.6	5.0 ± 1.2	11.2 ± 0.2	5.7 ± 0.5	0.58 [57]	0.15	0.27
**2**	26.1 ± 5.3	18.5 ± 0.3	16.0 ± 1.8	7.6 ± 0.2	17.2 ± 2.3	10.1 ± 1.0	0.27 [58]	0.28 ± 0.06	0.42 ± 0.02
**3a**	71.0 ± 9.5	59.1 ± 3.7	58.0 ± 4.0	34.8 ± 3.4	65.0 ± 2.8	24.4 ± 0.9	–	–	2.23 ± 0.07
**3b**	89.3 ± 8.0	83.0 ± 13.0	85.5 ± 8.3	83.1 ± 7.5	85.0 ± 0.1	74.2 ± 11.2	–	–	–
**3c**	86.1 ± 4.2	79.2 ± 7.6	95.3 ± 2.9	92.7 ± 3.5	99.1 ± 15.7	95.4 ± 4.0	–	–	–
**4**	18.8 ± 1.1	7.7 ± 3.3	6.0 ± 0.9	2.9 ± 0.2	8.0 ± 0.9	4.5 ± 0.2	1.11 ± 0.30 [34]	0.11 ± 0.04	0.21 ± 0.02
**5**	26.1 ± 3.5	16.3 ± 1.0	14.2 ± 2.0	9.1 ± 0.6	9.3 ± 2.6	3.4 ± 0.3	0.20 ± 0.08	0.05 ± 0.01	0.15 ± 0.08
**6a**	81.5 ± 8.0	68.4 ± 9.8	75.6 ± 5.9	49.6 ± 0.8	38.8 ± 3.1	19.0 ± 0.1	–	–	1.94 ± 0.16
**6b**	76.8 ± 1.9	74.3 ± 0.1	85.4 ± 0.6	71.4 ± 0.2	62.4 ± 8.3	46.4 ± 2.7	–	–	–
**6c**	89.9 ± 5.2	84.3 ± 10.4	90.2 ± 4.7	80.8 ± 1.9	95.2 ± 9.4	79.8 ± 8.6	–	–	–
CX-4945	0	0	0	0	2.3 ± 0.1	1.2 ± 0.1	–	–	–

**Table 2 ijms-22-06261-t002:** Viability of MCF-7 and MDA-MB-231 human breast carcinoma cells after the treatment with the tested compounds. The EC_50_ values were calculated using MTT-based assay and an equation Y = 100/(1 + 10^((LogIC_50_ − X) * HillSlope)).

Cpd.	EC_50_ ± SD [μM]						Log *P* *	Log *S* *
	MCF-7			MDA-MB-231				
	24 h	48 h	72 h	24 h	48 h	72 h		
**1**	20.83 ± 1.49	18.82 ± 1.60	16.86 ± 1.36	51.90 ± 14.43	25.11 ± 2.04	24.92 ± 0.25	4.33	−6.23
**2**	24.09 ± 1.39	20.08 ± 1.36	19.60 ± 1.68	54.62 ± 4.32	27.19 ± 0.49	20.57 ± 0.86	3.93	−5.37
**3a**	29.34 ± 1.24	23.05 ± 0.77	21.51 ± 1.86	40.80 ± 1.58	27.84 ± 1.31	24.52 ± 0.48	5.51	−6.55
**3b**	39.56 ± 1.06	28.38 ± 1.71	24.77 ± 1.33	95.58 ± 20.40	32.21 ± 3.26	27.30 ± 1.26	6.40	−7.49
**3c**	24.34 ± 0.25	21.14 ± 1.23	29.85 ± 1.45	45.11 ± 5.43	23.68 ± 3.97	32.77 ± 3.25	7.29	−8.43
**3d**	>100	>100	>100	>100	>100	>100	9.07	−10.28
**4**	4.12 ± 0.03	4.20 ± 0.29	2.66 ± 0.19	12.78 ± 6.60	5.35 ± 0.82	4.73 ± 0.48	4.46	−5.90
**5**	17.62 ± 6.35	15.96 ± 5.31	4.90 ± 2.71	57.84 ± 20.79	21.03 ± 9.85	9.99 ± 0.44	4.05	−5.04
**6a**	20.15 ± 1.74	18.07 ± 0.52	16.47 ± 1.13	52.17 ± 11.68	20.49 ± 1.97	16.85 ± 4.07	5.64	−6.21
**6b**	22.35 ± 1.12	18.59 ± 0.18	18.31 ± 2.86	51.43 ± 8.22	21.16 ± 2.40	22.06 ± 3.18	6.53	−7.15
**6c**	25.63 ± 12.30	20.46 ± 2.92	14.85 ± 2.14	61.28 ± 2.23	29.83 ± 9.85	21.30 ± 2.24	7.42	−8.08
**6d**	>100	>100	>100	>100	>100	>100	9.19	−9.92

* Calculator plugins were used for log*S* at 7.4 pH and log*P* calculations. MarvinSketch 14.9.1.0 (2014), ChemAxon (http://www.chemaxon.com) accessed on 30 August 2016.

## Data Availability

The data presented in this article are openly available.

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
