# Peer review of "Synthesis of Novel Acyl Derivatives of 3-(4,5,6,7-Tetrabromo-1H-benzimidazol-1-yl)propan-1-ols—Intracellular TBBi-Based CK2 Inhibitors with Proapoptotic Properties"

_ijms, 2021, doi:10.3390/ijms22126261_

Round 1

Reviewer 1 Report

Comments to the Author
The authors developed novel CK2 inhibitors and demonstrated their efficacy against breast cancer cell lines in this manuscript. The study was well designed and the concept of this study is interesting. However, there are some criticisms in the present study. 

  1. In the current study, the authors employed the breast cancer cell lines including MCF-7 and MDA-MB231. In my understanding, MDA-MB231 is the TNBC model cancer cell line, and MCF7 is the estrogen and progesterone receptors positive cell line representative for luminal-like breast cancer. The characteristics of the two cell lines are quite different from each other that may affect the efficacy of CK2 inhibitors. Why have the authors selected these cell lines in this study?
  2. I am curious about the differences in the time to induce apoptosis by CK2 inhibitors between two cell lines. The authors should explain the differences in the efficacy of novel CK2 inhibitors between MCF-7 and MDA-MB-231.
  3. In addition to antitumor activity, target selectivity is important for anti-cancer agents. How about the efficacy of novel CK2 inhibitors against a normal cell?
  4. It is important to check the dysregulation of CK2 in breast cancer cell lines employed in this study to investigate the efficacy of CK2 inhibitors. Have authors examined CK2 activity and/or CK2 mutation in MCF-7 and MDA-MB231?
  5. I recommend authors investigate the effect of CK2 siRNA against MCF-7 and MDA-MB231 to confirm that CK2 is a vital factor in these cell lines.
  6. How about the localization of CK2 in MCF-7 and MDA-MB231? Cytoplasm? Nucleus? CK2 inhibitors target the CK2 located in the cytoplasm?
  7. The authors examined the effect of novel CK2 inhibitors on cell proliferation and apoptosis using western blot and flow cytometry in this study. CK2 is also involved in DNA repair and regulation of the cell cycle. Have authors tested the effect of CK2 inhibitors on these functions of CK2?

Author Response

Response to the Reviewer’s #1 comments:

We are thankful for all the Reviewer’s comments and suggestions. Our point-by-point response is provided below.

  1. In the current study, the authors employed the breast cancer cell lines including MCF-7 and MDA-MB231. In my understanding, MDA-MB231 is the TNBC model cancer cell line, and MCF7 is the estrogen and progesterone receptors positive cell line representative for luminal-like breast cancer. The characteristics of the two cell lines are quite different from each other that may affect the efficacy of CK2 inhibitors. Why have the authors selected these cell lines in this study?

 Both cell lines differ in p53 status and CK2 level (Gray et al., 2014).

The following sentence was corrected by adding an additional information (in bold):

“In this context, we examined cytotoxic effects of all the obtained esters against two different human breast cancer cell lines (i.e., MCF-7 and MDA-MB-231), differing in p53 status and CK2 level [33].

  1. I am curious about the differences in the time to induce apoptosis by CK2 inhibitors between two cell lines. The authors should explain the differences in the efficacy of novel CK2 inhibitors between MCF-7 and MDA-MB-231.

The following sentence in discussion was added:

„The indicated differences in the efficacy of novel CK2 inhibitors between MCF-7 and MDA-MB-231 can be related to the caspase-3 deficiency in MCF-7 cells that consequently undergo cell death that lacked typical apoptotic properties [71].“

And the following sentence was corrected:

„Moreover, it should be also emphasized that the obtained results are promising regarding that both studied cell lines are difficult to enter into apoptotic pathway, since MDA-MB-231 cell line has status p53-/- what makes it less prone to enter apoptosis.“

  1. In addition to antitumor activity, target selectivity is important for anti-cancer agents. How about the efficacy of novel CK2 inhibitors against a normal cell?

We agree that such study could provide interesting information on the cytotoxic selectivity of the compounds. Unfortunately, due to the coronavirus pandemic, our studies had to undergo significant time-limitations, therefore we decided to focus exclusively on anticancer properties of the newly obtained compounds. However, we intend to continue these studies in a due course.

  1. It is important to check the dysregulation of CK2 in breast cancer cell lines employed in this study to investigate the efficacy of CK2 inhibitors. Have authors examined CK2 activity and/or CK2 mutation in MCF-7 and MDA-MB231?

Gray et al., 2014 (cited in manuscript as no 33) have studied both gene status and activity of CK2 in MCF-7 and MDA-MB-231, respectively. The authors concluded that both of these cell lines demonstrate heterozygous loss of CSNK2A2. MCF-7 cells additionally have a heterozygous loss of CSNK2A1, while MDA-MB-231 cells have gains on CSNK2B. They also compared for MDA-MB-231, MCF-7, MCF-10A (an immortalized breast epithelium line) and various normal murine tissues, and concluded that the three altered breast lines possess extraphysiological expression of CK2, even when compared with the brain (high level). In summary, the results obtained within this paper confirmed that CK2 expression is elevated in cancerous tissues. Moreover, Gray et al. demonstrated that MDA-MB-231 cells have much higher (>4 times) rate of CK2 kinase activity than MCF-7 cells.

  1. I recommend authors investigate the effect of CK2 siRNA against MCF-7 and MDA-MB231 to confirm that CK2 is a vital factor in these cell lines.

 We agree that investigating the effect of CK2 siRNA against MCF-7 and MDA-MB231 would be very informative. However, due to the logistics, we are unable to perform this study in the near future. However, it has been previously demonstrated that inhibition of CK2 by CX-4945 in the examined cell lines caused a range of phenotypic alterations, including decreased viability, cell cycle arrest, apoptosis and loss of migratory capacity, thus proving the tremendous potential of CK2 as a clinical target in breast cancer (Gray et al., 2014 cited in manuscript as no 33).

  1. How about the localization of CK2 in MCF-7 and MDA-MB231? Cytoplasm? Nucleus? CK2 inhibitors target the CK2 located in the cytoplasm?

It was demonstrated that in normal, quiescent, cultured fibroblasts, CK2α presents a predominantly cytoplasmic distribution. Both tested by us protein markers of CK2 activity (p-Akt Ser129 and p-p65 Ser529) are phosphorylated by CK2 in the cytosol, thus we can assume that tested compounds inhibit cytoplasmic CK2. However, translocation of CK2α from the cytosol to the nucleus during cell cycle progression after growth stimulation of serum-deprived quiescent cells was reported. Moreover, upregulation of nuclear CK2 protein levels has been observed in squamous cell carcinoma of the head and neck and breast carcinoma, both with poor clinical outcomes. It is then plausible that in our hands CK2 is mainly localized in the cytoplasm. Unfortunately, due a short period of time given us for the revision we are not able to perform immunolocalization studies.

  1. The authors examined the effect of novel CK2 inhibitors on cell proliferation and apoptosis using western blot and flow cytometry in this study. CK2 is also involved in DNA repair and regulation of the cell cycle. Have authors tested the effect of CK2 inhibitors on these functions of CK2

 The results were completed with the data from the cell cycle analysis in MCF-7 cells. Due to the limited time for the revision, we were not able to perform enoungh number of biological replicates, therefore, results were included in supplementary materials, as preliminary results (table S2 and Figure S5).

The following subsection has been added in the manuscirpt:

4.2.8. Detection of Cell Cycle Progression by Flow Cytometry

MCF-7 cells were cultured in 6-well plates and treated with the tested compounds used in concentrations corresponding to their EC50 values (72 h). After exposure to the compounds, the cells were trypsinized, collected and washed with cold PBS and fixed at –20°C in 70% ethanol for at least 24 h. Subsequently, then cells were washed in PBS and stained with 50 μg/ml PI (propidium iodide) and 100 μg/ml RNase solution in PBS sup-plemented with 0.1% v/v Triton X-100 for 30 min in the dark at the RT. Cellular DNA con-tent was determined by flow cytometry employing CyFlow Cube 8, Sysmex (Sysmex, Norderstedt, Germany) flow cytometer. The DNA histograms obtained were analyzed us-ing MacCycle software (Phoenix Flow Systems, San Diego, CA, USA) FCS Express 5 Flow software (De Novo Software, Glendale, CA, USA) for evaluation distribution of the cells in different phases of the cell cycle.

Reviewer 2 Report

Borowiecki et al reported the synthesis of a novel set of tetrabromo-benzotriazole derivatives aiming at improving the cellular activity or cellular permeability of previously discovered inhibitor of CK2 protein. To achieve this aim the authors has designed a set of 8 analogues by incorporating a hydroxypropyl moiety and it corresponding esters at position 3 of the tetrabromo-benzotriazole scaffold and to its 2-Me analogue. Although the synthetic route seemed to be straightforward, the authors have investigated several biocatalytic conditions for the esterification reaction. Next, the authors have evaluated the activity of synthesized compounds against CK2 and P1M1 kinases which led to identification of compd 5 and 2 with the most  inihibiton for CK2alpha (0.19uM and 0.27uM) and CK2 holoenzyme (60 nM and 028uM). These results indicated that the incorporated hydroxypropyl moiety has a positive impact on the enhancement of the inhibitory activity. Interestingly, non of the esterified compounds showed any inhibition potency. These results were further confirmed by the molecular docking study which showed that both of compd 5 and 2 have similar binding energy. However, the Me-substitution of compd 5 made the Ph-ring twisted and loss 2 hydrophlic interactions with Ser51 and Asp175. Then, the authors investigated the cellular activity of all synthesized compounds by performing a series of experiments to investigate their hypothesis regarding the cellular permiability and pro-apoptotic effect. Accordingly, compd 6b has been identified as the best pro-drug with potent anticancer activity.

Indeed, this is a great study which contains a huge series of well designed experiments for discovery or development of antineoplastic agents. The results are perfectly presented and all experimental details are included which clearly support the authors conclusion. I would recommend the publication of this study, after the authors consider the following concerns:

1- CX-4945 i well known to be potent CK2 inhibitor, however, in table 1 it showed no activity at all which is quite weird.

2- why the authors did not include CX-4945 in the cytotoxic evaluation as controller?

3- regarding the alkylation reaction, why the authors did not try to use NaH instead of DBU or K2CO3, it is clear that the basicity of N-is really low by the effect of 4Br? indeed 96h under reflux is too much!!

4- Regarding the authors hypothesis about 6b and 5 (even about the aim of this study), I think this hypothesis can be easily proved by just incubating the compound in cells and after 24h the authors can analyze by LC-MS the cell lysate to see if they can detect compd 5, then its done and proved.

5- regarding the in silico study, the authors should show the binding of the co-crystallised ligand and highlight the essential amino acid residues for efficient binding affinity. The authors should also show the binding of 6b. Regarding the binding of 5 and 2, as authors presented the authors showed that compd 5 lost 2 hydrophilic interactions by the tweisting of phenyl ring, which means that the affinity of 5 is lower than 2, the authors should highlight this point.

Author Response

Response to the Reviewer’s #2 comments:

  • CX-4945 is well known to be potent CK2 inhibitor, however, in table 1 it showed no activity at all which is quite weird.

We apologize for not explaining meaning of the values presented in Table 1. The residual activity of CK2 is presented at two different concentrations of the tested compounds, and 0% residual activity of CK2 means a 100% inhibition of the studied enzymes. Therefore, no residual CK2 activity for CX-4945 means that this compound entirely inhibits the kinase in the given concentration.

2- why the authors did not include CX-4945 in the cytotoxic evaluation as controller?

In the present study, we used TBBi as a cytotoxicity standard. We admit that this was performed by us deliberately as we wanted to show that all the applied structural modifications of the parental compound led to cytotoxic agents with improved anti-cancer properties when compared with TBBi. For our excuse, we can only add that we have previously tested CX-4945 on the same cell lines, and gathered a number of reliable data from MTT tests  that report on its EC50 values. These results have been already published elsewhere (i.e., WiĹ„ska et al., 2020, WiĹ„ska et al., 2018).

3- regarding the alkylation reaction, why the authors did not try to use NaH instead of DBU or K2CO3, it is clear that the basicity of N-is really low by the effect of 4Br? indeed 96h under reflux is too much!!

During our over-decade synthetic campaigns toward structural modifications of TBBi/TBBt inhibitors, we examined a lot of various alkylating systems to develop efficient substitution on N-atoms of parent molecules. These encompass the employment of almost all readily available bases, including: (i) hydroxides (NaOH, KOH, LiOH), (ii) carbonates (K2CO3, Na2CO3), (iii) organic (DBU) (iv) hydrides (NaH), (v) amides (NaNH2) and (vi) oxides (tert-BuOK) as well as (vii) organolithium reagents (n-BuLi, LDA) etc. All of them easily deprotonate very acidic hydrogen atom from N-H moiety due to swift colour and/or solubility changes of the reaction mixture after addition of the appropriate base. Nevertheless, it is more than obvious that the deprotonation itself is not the rate-limiting step in this case, but the substitution reaction, which might be dependent on stabilization of the formed anion, and in consequence, on its reactivity toward alkylating agents. When deprotonation of N-H acid is performed with alkali metal derivatives, a very strong ionic pair is probably formed and the nucleophilicity of the anion is decreased as it is hindered by the tightly bonded cation. This of course is also dependent on the solvent polarity and its potency to solvate ions in a selective manner. We admit not to test PTC conditions as well as additivies of crown ethers along with metal alkali bases; however, as so many experiments have been done in this issue, it cannot be ruled out so easily. This could be beneficial for the reactions‘ outcome, especially as the crown ethers are well-known to selectively complex Na+ and/or K+. Maybe the addition of these ethers would eliminate the formation of strong ionic pair, and deprotonated TBBi/TBBt would become more nucleophilic and thus reactive toward alkylating agents. In turn, we found DBU very efficient as base since it does not possess any counterion which could deactivate the desired anion, and probably this is its’ biggest advantage in such reactions.

4- Regarding the authors hypothesis about 6b and 5 (even about the aim of this study), I think this hypothesis can be easily proved by just incubating the compound in cells and after 24h the authors can analyze by LC-MS the cell lysate to see if they can detect compd 5, then its done and proved.

We are very thankful to Reviewer #2 for such a valuable advice, and definitely agree with her/him that it would be a direct proof of our hypothesis raised in this article. However, the development of reliable HPLC method coupled with MS is often troublesome since it requires a time-consuming optimization. Nevertheless, we surely extend our further studies toward suggested analyses.

5- regarding the in silico study, the authors should show the binding of the co-crystallised ligand and highlight the essential amino acid residues for efficient binding affinity. The authors should also show the binding of 6b. Regarding the binding of 5 and 2, as authors presented the authors showed that compd 5 lost 2 hydrophilic interactions by the tweisting of phenyl ring, which means that the affinity of 5 is lower than 2, the authors should highlight this point.

In silico study was performed additionally to our inhibition experiments. Possible binding modes of ligands 2 and 5 in the active site of human CK2α are presented in the Supplementary material (Figure S2). It was shown that hydroxyl group of 2 can form two polar interactions with key residues: Asp175 and Ser51. In the case of 5 one polar interaction with His160 was detected (loss of one polar interaction). However, calculated binding energy for both complexes are very similar (ΔGcalc -7.0 and -7.1). Finally, inhibition experiments revealed that both compounds inhibit CK2α with similar Ki values (0.27 and 0.20 μM respectively).

Considering very low inhibitory potency of 6b against CK2α (74.3% residual enzyme activity in presence of 10 μM 6b) we see no need for prediction of a possible 6b interaction with CK2.

Round 2

Reviewer 1 Report

The manuscript has been revised well. I think that the results are well discussed and conclusions are meaningful in the manuscript.

Reviewer 2 Report

Thanks to the authors for addressing all the points raised by the reviewer. Accordingly, I would recommend the publication of this study.